# Comparing the Effects of Differential and Visuo-Motor Training on Functional Performance, Biomechanical, and Psychological Factors in Athletes after ACL Reconstruction: A Randomized Controlled Trial

**DOI:** 10.3390/jcm12082845

**Published:** 2023-04-13

**Authors:** Fatemeh Gholami, Amir Letafatkar, Yousef Moghadas Tabrizi, Alli Gokeler, Giacomo Rossettini, Hadi Abbaszadeh Ghanati, Wolfgang Immanuel Schöllhorn

**Affiliations:** 1Department of Biomechanics and Sport Injury, Faculty of Physical Education and Sports Sciences, Kharazmi University, Tehran 1571914911, Iran; std_gholami_f@khu.ac.ir (F.G.); h.abbaszade3343@gmail.com (H.A.G.); 2Department of health and sport medicine, Faculty of Sport Sciences and health, University of Tehran, Tehran 1439813141, Iran; moghadas@ut.ac.ir; 3Exercise Science and Neuroscience, Department Exercise & Health, Faculty of Science, Paderborn University, 33098 Paderborn, Germany; 4Amsterdam Collaboration on Health & Safety in Sports, Department of Public and Occupational Health, Amsterdam Movement Sciences, Amsterdam UMC, 1105 Amsterdam, The Netherlands; 5School of Physiotherapy, University of Verona, 37134 Verona, Italy; giacomo.rossettini@gmail.com; 6Department for Training and Movement Science, Johannes Gutenberg University Mainz, 55122 Mainz, Germany

**Keywords:** anterior cruciate ligament reconstruction, rehabilitation, motor learning, differential learning, visuo motor learning

## Abstract

Variation during practice is widely accepted to be advantageous for motor learning and is, therefore, a valuable strategy to effectively reduce high-risk landing mechanics and prevent primary anterior cruciate ligament (ACL) injury. Few attempts have examined the specific effects of variable training in athletes who have undergone ACL reconstruction. Thereby, it is still unclear to what extent the variations in different sensor areas lead to different effects. Accordingly, we compared the effects of versatile movement variations (DL) with variations of movements with emphasis on disrupting visual information (VMT) in athletes who had undergone ACL reconstruction. Forty-five interceptive sports athletes after ACL reconstruction were randomly allocated to a DL group (n = 15), VT group (n = 15), or control group (n = 15). The primary outcome was functional performance (Triple Hop Test). The secondary outcomes included dynamic balance (Star Excursion Balance Test (SEBT)), biomechanics during single-leg drop-landing task hip flexion (HF), knee flexion (KF), ankle dorsiflexion (AD), knee valgus (KV), and vertical ground reaction force (VGRF), and kinesiophobia (Tampa Scale of Kinesiophobia (TSK)) assessed before and after the 8 weeks of interventions. Data were analyzed by means of 3 × 2 repeated measures ANOVA followed by post hoc comparison (Bonferroni) at the significance level of *p* ≤ 0.05. Significant group × time interaction effects, main effect of time, and main effect of group were found for the triple hop test and all eight directions, SEBT, HF, KF, AD, KV, VGRF, and TSK. There was no significant main effect of group in the HF and triple hop test. Additionally, significant differences in the triple hop test and the seven directions of SEBT, HF, KF, KV, VGRF, and TSK were found between the control group and the DL and VMT groups. Between group differences in AD and the medial direction of SEBT were not significant. Additionally, there were no significant differences between VMT and the control group in the triple hop test and HF variables. Both motor learning (DL and VMT) programs improved outcomes in patients after ACL reconstruction. The findings suggest that DL and VMT training programs lead to comparable improvements in rehabilitation.

## 1. Introduction

Despite significant progressions in surgical procedures and suggestions to optimize rehabilitation, the short- and long-term outcomes after ACL reconstruction remain disappointing [1]. The risk of sustaining a second ACL injury is nearly one in four in athletes younger than 25 years returning to high-risk sports activity [2]. In addition, an ACL injury and reconstruction are unequivocally associated with the development of knee osteoarthritis [3]. Moreover, an ACL injury can have detrimental psychological effects on the athlete as well (self-efficacy, fear of movement/re-injury) [4].

Asymmetries in leg coordination are commonly observed during daily and sport activities after an ACL injury and/or following ACL rupture (ACLR) [5,6,7,8,9]. Unfortunately, contemporary rehabilitation plans do not effectively target dysfunctional movement patterns and motor control [10]. Dysfunctional movement control has been linked with an augmented risk for ipsilateral or contralateral secondary injury and the evolution of premature development of knee osteoarthritis [11,12]. Based on these continued neuromuscular control deficits, it is apparent that traditional rehabilitation, which most often relies on predefined and person-independent movement prototypes connected with numerous error-corrections, does not restore normal motor function in all patients after ACLR [13].

In contrast, rehabilitation approaches that rely on self-organisation with a strong emphasis on the individuality of movements have been proposed more recently [13,14,15]. These approaches pursue a more holistic strategy that is based on general principles of motor learning, system dynamics, and neurophysiology, relies on the offer of increased noise in terms of more variations [15], and thereby fosters implicit learning [16], such as differential learning (DL) [17,18]. Both general motor learning approaches have in common that they try to reduce the dominance of the working memory [19,20] and thus, the activation of the frontal lobe [21]. While the implicit learning approach relies on the modification of instructions and staying below the capacity threshold of the working memory, the DL intends to overload the working memory by adding stochastic perturbations [22] or increased noise to the to-be-learned movement to trigger a qualitative change of frontal lobe activation [23,24]. A more specific approach, thus far mainly related to rehabilitation after ACLR, that focuses more on visual perturbations is suggested with the visuo-motor training approach [25]. 

In more detail, the DL approach models the patient or learner using the versatile stimulations of the action-apperception system through increased fluctuations in the surrounding of the movement to-be-learned to make the system instable. In its most extreme form, no repetition and no augmented feedback are given to the learner. By not giving the learner explicit information about a possible solution, a true self-organization process is initiated [25]. The increased noise, transmitted mainly by means of mechano-sensors (e.g., Vater–Pacini, Merkel cells, Golgi, Ruffini organs, muscle spindles), vestibular organs, and the visual system, leads to a broader spectrum of input signals to the neural networks of the different brain areas, allowing easier discovery of new and more effective activation and movement patterns [18,26]. Preference is given to the proprioceptive systems during the learning of a movement technique due to the highly parallel processing outside of visual and conscious control. This course of holistic action is in analogy with the noisy training of artificial neural networks for higher learning rates, whose principal working mechanisms were originally derived from the behavior of neurons [27]. As an alternative to the pedagogically enticing [28] but epistemologically problematic [29] constraints model, the measures that trigger the increased noise of interventions in DL are differentiated into internal and external. Due to the different time scales of adaptation, the internal measures are further distinguished into metabolism- and emotion-related measures (e.g., varying fatigue and mood) on one side, and cognition-oriented measures (e.g., problems to be solved) on the other side. The external measures are associated with varying equipment, clothes, landscape, obstacles, etc. This structure goes along with the differentiation of objective (external) and subjective (internal) information proposed from cybernetic pedagogy [25,30], and emphasizes the closer interconnection of the two internal subsystems. In addition to numerous studies in the field of sports [18], there have been increasingly positive findings in the field of rehabilitation [31], e.g., stroke patients [32], focal dystonia [33], and hip replacement [34], but no study on the rehabilitation process after ACLR has been conducted thus far.

Alternatively, the visuo-motor training (VMT) assumes a “more accurate feedforward motor control” by stronger emphasis on visuo-motor information during practice [35,36,37]. In a more specific context, the VMT was most recently proposed as a potential avenue to augment ACL rehabilitation to facilitate sensory reweighting (nervous system adjustment of relative sensory input/processing for motor control) by shifting the post-injury reliance stronger exclusion of the visual part during motor control to the remaining proprioceptive inputs, namely, the joint capsule, other ligaments, and muscle spindles [38]. In particular, the use of visual obstruction training aimed at sensitizing the esthetic and visual input during standard rehabilitation exercises [38]. VMT intends to reduce the dependence on vision by shifting neural processing towards proprioception and increasing the efficiency of visual processing [39]. This is mainly derived theoretically from studies suggesting neurophysiological changes in athletes after ACLR that include: (a) modifying visual input combined with altered sensorimotor processing, which may induce (b) increased visual and somatosensory processing to plan movement and maintain neuromuscular control, and (c) increased cortical top-down motor control strategies [39]. 

To date, there is a paucity of studies comparing DL and VMT in patients after ACLR. Accordingly, this study aimed to compare the effect of the DL and VMT compared to a control group with traditional treatment on functional performance, biomechanical, and psychological factors in athletes with a history of ACL reconstruction. We hypothesized that: (a) both intervention groups have superiority over the control group, and (b) that DL and VMT should achieve comparable improvements.

## 2. Materials and Methods

### 2.1. Study Design

This study was a single monocentric assessor blind randomized controlled trial (RCT) prepared and reported following CONsolidated Standards of Reporting Trials (CONSORT) guidelines [40], and all intervention conditions conformed to the Declaration of Helsinki [41]. This RCT was approved by the Ethics Committee of (blinded for submission) and retrospectively registered at UMIN-CTR (ID number: UMIN000047952). All patients provided written informed consent prior to enrollment.

### 2.2. Participants Recruitment and Eligibility Criteria 

Forty-five competitive male handball, volleyball, and basketball players with a primary ACL reconstruction, who completed conventional post-operative rehabilitation, voluntarily enrolled in a group training program. After assessment of the study’s criteria, they were randomly assigned to one of three groups: DL group (n = 15), VMT (n = 15), and control group (n = 15). The study was conducted in the biomechanics lab of the Kharazmi University, Iran, from March of 2020 to July of 2020. Athletes were enrolled through one of the authors, while a blinded investigator was responsible for the randomization.

The following eligibility criteria were applied:

Inclusion criteria were: Having a unilateral hamstring tendon autograft ACL reconstruction performed by the same surgeon 6–12 months prior to participation, athletes were required to exhibit no pain, no effusion, report pain-free knee active range of motion (via electro goniometer), achieve 80% or greater quadriceps strength index [42], limb symmetry via a handle-held dynamometer, and complete all hop tests without pain and at an equivalent distance/rate of at least 80% of the contralateral limb. Moreover, they were cleared by their medical teams to resume sports participation [43,44].

Exclusion criteria were having a concomitant injury to another knee structure (e.g., medial collateral ligament, meniscus) or experiencing a post-operative re-injury [43].

### 2.3. Demographic and Health Data

At baseline, there were no significant differences (*p* > 0.05) in demographic data among the three intervention groups (Table 1) (age (*p* = 0.673), mass (*p* = 0.566), height (*p* = 0.710), Body Mass Index (*p* = 0.649), time since surgery (*p* = 0.392)). The same was valid between the groups at baseline for any of the dependent variables of interest (*p* ≥ 0.100), indicating that the groups were comparable in terms of initial anthropometry and injury history.

### 2.4. Randomization, Allocation, and Implementation

Concealed allocation was performed using a computer-generated http://randomizer.org/ (Social Psychology Network, middeltown, CT, USA, accessed on 22 June 2013) block randomized table of numbers (1, for the control group; 2, for the DL group; and 3, for the VMT group). The random numerical sequence was placed in sealed opaque envelopes in a box. According to the group assignment, another researcher (blinded to the baseline assessment) opened envelopes and proceeded with training. 

### 2.5. Sample Size Calculation 

A sample size estimate indicated that 15 athletes per group (15 total athletes) would provide adequate statistical power to detect a group-by-time interaction for moderate effect size (partial eta squared = 0.06) [45]. This determination was made based on biomechanical and joint position sense data [46,47,48]. Using these data, an alpha of 0.05, a beta of 0.20, the effect size of ηp^2^ = 0.06, and assuming a correlation among repeated measures of 0.85 for our sample size estimate, we arrived at the total of 45 athletes needed. The value used for the correlation among repeated measures was based on the test-retest reliability reported for isokinetic testing. G*Power software was used for sample size estimation [17].

All athletes in the experimental groups participated in each scheduled training session (100% compliance). In addition, all athletes who completed baseline testing also returned for follow-up testing (Figure 1). 

### 2.6. Procedures

Before participating, a licensed athletic trainer with five years of experience conducted a preliminary assessment (based on inclusion and exclusion criteria) to ensure that participation was safe for all included athletes. 

Baseline and post-intervention were completed using two blinded assessors before and after the interventions. Athletes in the control group also completed baseline and post-intervention and performed their typical training regimen, such as improving technique and sport-related skills over a similar period. The primary outcome was functional performance (triple hop test), while secondary outcomes were dynamic balance during Star Excursion Balance Test (SERT), biomechanics during single leg drop jump (knee flexion (KF), ankle dorsiflexion (AD), knee valgus (KV), and vertical ground reaction force (VGRF)), and kinesiophobia. 

Athletes executed a standardized warm-up protocol, including double-leg squats (2 × 8 repetitions) and double-leg maximum jumps (2 × 5 repetitions), followed by calf-stretching with a straight and bent knee [49]. In addition, all athletes were asked to refrain from training, maintain a regular diet, and avoid smoking, caffeine, and alcohol for 24 h prior to testing sessions [49].

Each test was performed on the non-involved side and repeated on the involved side, except for descent to fatigue. For this test, the initial limb test to control the potential fatigue of the hip and core muscles may affect the function of the second limb. 

## 3. Primary Outcome Measure

### Functional Performance (Triple Hop Test)

The athletes completed three trials of a single-leg, horizontal, triple hop test. Athletes performed practice trials to become familiar with the task prior to testing. A standard tape positioned perpendicular to the starting line was used to measure the distance hopped. The single-leg triple hop task is a common and reliable (ICC = 0.93–0.98) functional performance task used for athletes with ACL reconstruction [50]. 

## 4. Secondary Outcome Measures 

### 4.1. Dynamic Balance (Star Excursion Balance Test)

The Star Excursion Balance Test (SEBT) as a reliable tool (ICC = 0.90) was used for measuring dynamic balance [51]. Athletes were asked to stand in the center of eight lines forming an eight-pointed star, reach the farthest possible direction with their non-stance limb while maintaining balance on one leg, and return to the center of the grid. Athletes were also told to keep their hands on their hips to standardize torso and limb movements. If the athletes removed their hands from their hips, lifted the stance foot from the floor, lost balance during the trial, or the toe-touch was heavy or prolonged, the trial was considered an error and repeated. The reach distance was normalized and expressed as a percentage of the maximum reaching leg’s length. Each direction was tested for three trials with 30 s rest between each trial and a one-minute break between each direction. The represented SEBT for each direction was an average of the three trials [51,52].

### 4.2. Biomechanics during Single-Leg Drop-Landing Task 

For the single-legged drop-landing task, athletes were asked to assume a single-injured legged standing position on a platform of 30.5 cm height placed next to the force plate. Athletes landed on a force plate with the same limb and then jumped upward as high as possible. Each athlete was allowed to try the landing task 3 times. Three trials were collected for each participant [53]. The mean of these three single-legged drop-landings was adopted for statistical analysis. No augmented feedback was given during data collection.

### 4.3. Data Collection 

Kinematic data were recorded at 250 Hz using a six-camera Motion Analysis system (raptor E with associated Cortex software). Kinetic data were collected at 1500 Hz using an AMTI force plate (AMTI, Watertown, MA, USA) synchronized with the motion capture system. Retroreflective markers were placed on anatomic landmarks of the pelvis and lower extremities following the Plug-in-Gait lower body marker set. A static calibration trial was conducted while athletes stood in the anatomical position. Following the static calibration trial, the athletes completed a standardized warm-up (running and jumping tasks).

Kinematic and kinetic data from the single-leg landing trials were low pass filtered using a fourth order, zero-lag, recursive Butterworth filter. A cutoff frequency of 15 Hz was used for the marker data, and a cutoff frequency of 50 Hz was adopted for the force data. Three-dimensional joint angles were calculated for the trunk, hip, and knee using a XYZ Cardan sequence, which resulted in joint angles corresponding with flexion/extension, adduction/abduction, and internal/external rotation ((knee flexion (KF), ankle dorsiflexion (AD), knee valgus (KV)). Joint angles reflected the orientation of the local coordinate system of the distal segment relative to the local coordinate system of the proximal segment. All kinetic variables were identified during the first 100 ms following initial contact with the force plate. Loading rates were calculated by dividing the peak z-component of GRF by the time to peak force. All kinetic variables were normalized to body weight (BW) as appropriate. All data processing was performed using custom MATLAB scripts (The MathWorks, Inc., Natick, MA, USA) to extract kinetic and kinematics data during the initial landing phase of the single-legged landing task. The three trial mean was calculated for each of these aforementioned dependent variables.

### 4.4. Kinesiophobia (TSK)

Kinesiophobia was assessed using the Iranian version of the Tampa Scale of Kinesiophobia (TSK). The TSK has 11 items, with scores ranging from 11 to 44 points, and higher scores indicate greater pain-related fear of movement/re-injury. Therefore, the TSK-11 is a valid measure (ICC = 0.81) of fear of movement/re-injury in the later stages of rehabilitation after ACL reconstruction [45,54].

## 5. Interventions

### 5.1. Differential Learning (DL)

Participants in the DL group executed their exercises for 8 weeks under the supervision of an experienced athletic trainer. Each week comprised three sessions of 25–30 min (odd days). The exercises were performed under different conditions (Table 2), including exercises in the dark, on the sand with shoes and without shoes, and with loud music [13]. Additionally, variations of the double-legged jump were included, e.g., double legged jump on a Bosu-ball, double-legged jump over an obstacle, and double-legged jump with an air target. 

### 5.2. Visual-Motor Training (VMT)

Athletes in the VMT group also attended an 8-week training period, consisting of three sessions per week (even days), with every training session lasting approximately 25–30 min. VMT protocols were developed for stroboscopic intervention and integrated into regular training (Table 3) [55]. The VMT protocols consisted of tasks including a Tap-test, agility ladder drills, single-leg stance (on foam), vertical jumps, and squat jumps. In addition, an error scoring system with detailed criteria to assess behavioral performance while wearing stereoscopic glass (SG) is described in Table 4.

For VMT, athletes performed the protocol under stroboscopic conditions. Each week, the shutter glasses’ settings (frequency [Hz] and duty cycle [%]) were adjusted to compensate for expected improvements in visuomotor performance and to avoid adaptation to a specific setting [55]. Moreover, athletes should refrain from looking at their feet and focusing on how to exercise while performing the movements.

### 5.3. Control

Athletes in the control group received no specific treatment and were encouraged to continue their typical training regimen, such as improving technique and sport-related skills.

### 5.4. Statistical Analysis

The Shapiro–Wilk and Levene’s tests were conducted to evaluate the normality and homogeneity assumptions, and all the tested variables showed *p* > 0.05. Group demographics were compared using a one-way analysis of variance (ANOVA). To determine differences between the three groups and time (pretest and posttest) 3 × 2 repeated measures ANOVA was directed followed by post hoc comparison with Bonferroni correction [56]. Within group factor (pretest to posttest) as a main effect of time, and between group factor as a main effect of group were considered. Additionally, 95% confidence intervals (CI95%) were computed based on the adjusted group mean differences, and Cohen’s d effect size (ES) of 0.8, 0.5, and 0.2 were considered “large”, “moderate”, and “small” [57]. Statistical significance was set at *p* ≤ 0.05. All data analyses were calculated by means of SPSS software version 22.

## 6. Results

### 6.1. Functional Performance 

The triple hop test revealed a statistically significant group × time interaction (F2,42 = 3.861; *p* = 0.029) and main effect of time (F2,42 = 16.226; *p* = 0.001). There was only a trend for a significant main effect of group (F2,42 = 2.609; *p* = 0,085). Statistically significant differences in the triple hop test distance were found between the DL and control (*p* = 0.017, ES = 1.18) groups. A post hoc test showed that the DL (*p* = 0.001, ES = 1.15) and VMT groups (*p* = 0.003, ES = 0.90) exhibited a significantly larger improvement than the control group (Table 5).

### 6.2. Dynamic Balance 

Significant group × time interaction effects, main effect of time and main effect of group were found for the eight directions (*p* < 0.05). There was no significant between group differences in medial direction. At anterior (*p* = 0.001, ES = 3.74), antero-medial (*p* = 0.001, ES = 1.90), postero-medial (*p* = 0.001, ES = 2.70), posterior (*p* = 0.016, ES = 2.13), posterolateral (*p* = 0.001, ES = 2.75), lateral (*p* = 0.001, ES = 2.92), and antero-lateral (*p* = 0.001, ES = 2.64) directions significant differences were found between DL and control groups. Additionally, significant differences in anterior (*p* = 0.001, ES = 3.71), antero-medial (*p* = 0.001, ES = 1.87), postero-medial (*p* = 0.001, ES = 2.88), posterior (*p* = 0.032, ES = 2.05), postero-lateral (*p* = 0.002, ES = 2.45), lateral (*p* = 0.008, ES = 2.64), and antero-lateral (*p* = 0.001, ES = 2.35) directions were found between the VMT and control groups. Differences between intervention groups were not statistically significant. Post hoc test showed that the DL and VMT groups have significant larger improvements in anterior (*p* = 0.001, ES = 2.90; *p* = 0.001, ES = 3.40), anteromedial (*p* = 0.001, ES = 1.94; *p* = 0.001, ES = 1.86), medial (*p* = 0.001, ES = 1.94; *p* = 0.001, ES = 1.86), posteromedial (*p* = 0.001, ES = 2.70; *p* = 0.001, ES = 2.88), posterior (*p* = 0.001, ES = 2.39; *p* = 0.001, ES = 2.22), posterolateral (*p* = 0.001, ES = 2.92; *p* = 0.001, ES = 2.53), lateral (*p* = 0.001, ES = 2.78; *p* = 0.001, ES = 2.68), and anterolateral (*p* = 0.001, ES = 2.90; *p* = 0.001, ES = 2.37) directions (Table 6).

### 6.3. Biomechanics 

Significant group × time interaction effects, main effect of time, and main effect of group were found for the hip and knee flexion, ankle dorsiflexion, knee valgus, and VGRF (*p* < 0.05). The main effect of group was not significant at hip flexion angle. Significant differences in hip (*p* = 0.001, ES = 1.13) and knee (*p* = 0.001, ES = 2.70) flexion, knee valgus (*p* = 0.001, ES = 4.74), and VGRF (*p* = 0.001, ES = −2.05) were found between DL and control groups. Additionally, at knee flexion (*p* = 0.001, ES = 3.74), knee valgus (*p* = 0.001, ES = 3.67), and VGRF (*p* = 0.001, ES = −2.40), significant differences were found between the VMT and control groups. Differences between intervention groups (*p* > 0.05) were not statistically significant. Post hoc test showed that the DL and VMT groups exhibited a significant increase in hip (*p* = 0.001, ES = 1.05; *p* = 0.001, ES = 0.78) and knee (*p* = 0.001, ES = 1.86; *p* = 0.001, ES = 2.95) flexion and ankle dorsiflexion (*p* = 0.001, ES = 1.68; *p* = 0.001, ES = 1.90) angles and a significant decrease in knee valgus (*p* = 0.001, ES = 6.5; *p* = 0.001, ES = 4.24) and VGRF (*p* = 0.001, ES = −3.16; *p* = 0.001, ES = −1.02) (Table 7).

### 6.4. Kinesiophobia 

A statistically significant group × time interaction (F2,42 = 6.154; *p* = 0.001), main effect of time (F2,42 = 50.047; *p* = 0.001), and main effect of group (F2,42 = 3.438; *p* = 0.029) was reported for the TSK test (*p* < 0.001) (Table 5). A significant decrease in the TSK test was found between the DL and control (*p* = 0.001, ES = −2.59) groups and between the VMT and control groups (*p* = 0.001, ES = −2.59). Differences between the DL and the VMT groups were not statistically significant. Post hoc test showed that the DL (*p* = 0.001, ES = −2.42) and VMT groups (*p* = 0.003, ES = −2.64) exhibited a significant decrease in the TSK test (Table 8).

## 7. Discussion

Both experimental groups led to statistically significant improvements with large effect sizes of the selected performance variables after 8 weeks’ intervention, whereas the control group did not show any statistically significant changes. The study also revealed no statistically significant differences between the VMT and DL groups for all outcomes measured, such as functional performance (triple hop test), dynamic balance (SEBT), biomechanics (hip flexion, knee flexion, ankle dorsiflexion, knee valgus, and VGRF), and kinesiophobia (TSK). Nonetheless, the DL group in majority showed higher performance increases than the VMT group.

Both DL and VMT groups showed a significant increase from pre- to posttest in the triple hop test, with large effect size (*p* = 0.001, ES = 1.15; *p* = 0.003, ES = 0.90), and the DL illustrated a significant difference compared to the control group (*p* = 0.017, ES = 1.18). Therefore, according to the higher effect size on the side of the DL group (ES = 1.15 vs. 0.90), the DL group appears to lead to even better functional performance than the VMT group. Whether these differences become even bigger and more significant with a longer duration of intervention [58] needs further research. With the DL method, instead of applying repetitive augmented feedback by the therapist, athletes receive the most versatile internal and external feedback from their sensory systems through the information provided by the variation in successive movements caused by changes in every trial [59,60]. In addition, by not correcting the athletes, the psychological stress in the form of self-criticism and the critical comparison with previous trials for error detection becomes a factor. With this, a higher activation of the prefrontal lobe towards detrimental frequency bands can be assumed as well. DL training, based on variable practice, not only allows athletes to explore and choose more appropriate solutions according to the boundary conditions given by the external and internal situation, but leads to an increase in the adaptation of the individual to the situation as well [61]. More variety and increased variability during training sessions are considered functional, thus increasing the coordination set of individual movements and adaptation to the dynamically changing conditions [62]. It appears that the greater improvement in triple-hop test scores in the DL group is likely due to greater variability in training, subconscious knowledge, and experience regarding how to handle deviations from expected results. Regarding the functional performance, measured by means of flexion angles of the lower extremities, athletes from both intervention groups landed on the involved limb by maintaining a more extended knee position accompanied by more hip flexion and anterior pelvic tilt. It seems that the athletes needed to adopt this positioning of the entire kinetic chain as a compensatory mechanism for the reduced knee work found in all phases of the triple-hop task [63]. Specifically, the biomechanical analysis revealed alterations in the lower extremities. Generally, it is assumed that reduced hip, ankle, and knee flexion, as well as increased knee valgus, may increase the risk of ACL injury [64,65,66,67]. All of them are indicators for reduced stiffness in the lower extremities. The increased angles in these variables provide evidence for an increased stiffness by means of both intervention groups, which could reduce these originally hazardous joint positions. This may also have led to improvements in the triple-hop test. In fact, improvement in the performance of a triple hop masked significant lower limb deficits, especially in knee joint biomechanics in athletes after ACL reconstructions [63].

Regarding the dynamic balance, the DL and VMT groups demonstrated significant within group differences for all directions. Additionally, a significant increase between the control group compared to the DL and VMT groups for all directions were found, except for the medial direction. The directions of the performance of SEBT serve to detect bilateral neuromuscular control deficits [68]. Therefore, improving SEBT in both intervention groups of our study provides evidence for the effectiveness of the applied programs in neuromuscular control after ACL reconstructions. Given the high effect sizes reported, athletes in the DL group showed more improvements than the VMT group (except anterior and posteromedial directions). However, the differences between both did not achieve statistical significance. How this behaves over an even longer period of intervention would have to be specifically investigated [58]. These changes can be associated with the characteristics of the interventions. On the one hand, DL training supports the learner to become more agile and to be able to adopt to various boundary conditions in a shorter time more adequately [69]. On a physiological level, DL trains proprioception and kinesthetics in so many ways and implicitly, since the majority of movements already occur outside the field of view [69]. Whether this process could be supported by additional stroboscopic goggles or whether both approaches are mainly associated with a comparable change in prefrontal brain activation [70,71] that supports motor learning in general requires further research. On the other hand, in VMT, athletes were frequently asked to close their eyes while doing their variable motor training. In this context, it was hypothesized that closed eyes and perturbed vision can trigger increased proprioceptive training and improve sport-specific behavioral performance and aspects of neuro-cognition such as visual memory, anticipatory timing of movements, and central visual field motion sensitivity, leading to transient attention ability [72]. In fact, the motoric variations were trained similar to DL, but in comparison to DL, the motoric variations were more blocked and more reduced, while on the visual side, they were increased. The extent to which exclusively perturbing vision caused uncertainty and fear of re-injury requires further research. 

Regarding the biomechanical variables of the lower extremities, both VMT and DL training led to increased hip and knee extension and ankle dorsiflexion, decreased knee valgus, and decreased VGRF compared to control athletes. A previous study mainly reported the VMT interventions’ influence on biomechanical measures (e.g., knee sagittal- and frontal-plane excursions, peak moments, and vertical GRF) [72]. Similar to our results, Grooms et al. recently found comparable evidence due to a stroboscopic visual-feedback disruption that could alter the kinematics of sagittal- and frontal-plane landing knee, but did not significantly alter the knee joint moments. They stated that visual-motor ability might contribute to neuromuscular knee control [72]. Early research with SG explored behavioral performance on motion coherence, divided attention, multiple-object tracking [73], short-term visual memory [74], and anticipation [75], as well as performance on sports-specific tasks from single-leg squatting [76], ice hockey [77], tennis, and badminton [78]. These authors concluded visual perturbation training can improve sport-specific behavioral performance and aspects of neurocognition including visual memory, anticipatory timing of moving visual stimuli, and central visual field motion sensitivity and transient attention ability. However, the major aims of all the studied activities were related to target movements, which are highly dependent on the visual system. Furthermore, because of the lack of the comparison with another intervention group the part of the obscuring visual content in comparison to the exclusively proprioceptive aspect was missing. 

In the current study, a statistically significant decrease in the TSK test was found between the control group and the DL and VMT groups. Both groups showed pre- to posttest significant decrease. In addition, regarding kinesiophobia, patients with high fear of re-injury were identified during the rehabilitation process using a clinical questionnaire, such as the TSK [79]. Once individuals with high fear are identified, interventions such as goal setting can be implemented to improve outcomes [79]. In addition, movement patterns during functional tasks should be evaluated, and deficits or abnormal movements should be addressed during rehabilitation. Therefore, these means may support the return to sport or activity and reduce future injury risk [80,81,82]. Utilizing the sport injury risk profile promotes consideration for the sociocultural influences (e.g., coach/team RTS time expectations), mixed psychological states (e.g., fear of reinjury), and acknowledgement of shifted athlete goals throughout the recovery process. The athlete should also process the confounding neurocognitive and environmental components of RTS (i.e., weather, fan/opponent reactions, altered decision making in sport). It is well established that neurocognition and emotions can influence adherence to rehabilitation programs [83]. Adherence is a crucial component to successful recovery. With that in mind, it is recommended to consider the multitude of psychosocial factors the athlete with ACLR must navigate during the rehabilitation process to maximize rehabilitation outcomes [84].

Supporting the VMT approach, more recently, researchers have aimed to evaluate if neurocognitive processing (e.g., reaction time, processing speed, and visual-spatial memory) during computerized assessments is related to lower extremity injury risk and injury risk biomechanics. Healthy individuals with lower neurocognitive performance demonstrated higher injury-risk in jumping and cutting tasks [82,83]. Additionally, lower baseline neurocognitive performance has been retrospectively associated with increased risk of ACL injury occurrence [85]. Although further evidence is needed to understand the detailed relationships between various neurocognitive processes and lower extremity injury risk, the available evidence only partially suggests consideration for integration of neurocognitive interventions to rehabilitation from lower extremity musculoskeletal injury [86]. If one compares the time within which lower extremity injuries occur (<50 ms) with the time needed to consciously influence a stimulus coming from the lower extremity (>200 ms), the large difference alone shows the problems of a cognitive influence on an injury process. Just here, the DL already provides evidence that the effects of the interventions are in the time domain where injuries also occur [70,87]. Regardless of these first indications, more research will be needed to get a change in thinking in this direction.

The use of VMT, which aims to better integrate the influence of visual information by obscuring visual input during standard rehabilitative exercises, may reduce the dependency on vision and/or increase visuomotor processing efficiency [38]. With this, VMT suggests an alternative to the most common physical therapy following ACL injury, which emphasizes visual attention to the knee, as clinicians primarily utilize visually dominated exercises and provide feedback with an internal focus of attention (i.e., emphasizing the concentration on movement kinematics or muscle activation, rather than movement actions) to the injured joint [13,88,89,90,91]. Along with the research on the different effects of internal vs. external focus [92] this strategy needs to be rethought. Especially for athletes, this strategy with an internal focus may be maladaptive, when returning to a competitive sport environment, where the high demand to integrate dynamic visual information may limit the capacity of working memory to allocate neural resources to guide movement. Thereby, it is important to remember that the working memory model was originally derived from phenomena exclusively associated with serial, spatial-visual tasks [93] that were mainly studied in movements with a small number of degrees of freedom. More recently, Baddeley himself has emphasized that this model does not allow to generalize to dominant proprioceptive, kinesthetic, or somatosensory tasks [94] These tasks, in majority, are highly parallel, high-dimensional, and emphasize other sensory systems than the visual. Despite this lack of evidence and despite knowledge about the different central nervous processing of visual and proprioceptive information, etc., inadmissible generalized recommendations for motor learning have been derived [25].

In contrast to VMT, DL offers interventions addressing the real performance setting and development of skills and techniques through the continuous manipulation of specific [16] internal and external variations that are individual and situated. As a complex system that is highly sensitive to its initial conditions and, therefore, is not predictable, it enables the athlete not only to act adequately in constantly changing external conditions, but also to adapt to the ever-changing emotions and metabolic processes within [25] to solve a given movement problem in a real sport situation [16]. Initial studies already demonstrate a dual effect of DL training in high-performance sports. DL training applied to a female Austrian first division volleyball team during the season resulted in higher jumping performance over a longer period of time [60], in addition to improved balance performance [60], which is associated with preventive effects.

Variable instructions, as given in DL training, increase the probability to effectively convey goal-related information, and educators commonly use them to teach and refine motor performance at all levels of skill [95]. In contrast, some ACL injury prevention programs use discrete instructions guided by presumed correct movement execution and explicit rules for desired landing position by emphasizing proper hip, knee, and ankle alignment. For example, the main goal of the neuromuscular training program of Holm et al. was “to improve awareness and knee control during standing, cutting, jumping, and landing” [96]. The players were encouraged to focus on the quality of their movements with emphasis on the knee over-toe position [41]. This may be a commonly used approach, but the use of explicit strategies promotes the likelihood of fear of failure [20] triggers adverse comparisons that limit working memory, and, as a result, may be less appropriate for acquiring mastery of complex motor skills [97]. Instructions that direct performers’ attention to his or her own movements can actually have a detrimental effect on performance and learning and disrupt the execution of automatic skills, particularly in comparison with an externally directed attentional focus [92,98,99]. Therefore, we emphasize that an automized landing technique without too much explicit thinking about the correctness or incorrectness of positioning after a jump is much more advantageous for recovery and for prevention.

While VMT has its origins primarily in concrete physical therapy practice [35], and successively integrates neurophysiological findings, DL was derived from its inception from the much more general theories of dynamical systems [100,101,102] and early findings on neuroplasticity [103,104]. Since its transfer from motor developmental phenomena [105] and small motor cyclic movement forms [100] to large motor ballistic movement forms [102,104], Dynamic Systems theory has been accepted as a framework for numerous phenomena in movement research and is used as an explanatory approach for variations in movement performance in a wide variety of domains. Against the background of the emergence of both approaches, the VMT can be considered as a subset of DL, whereby the later goes far beyond the variation of visual aspects with corresponding effects. According to dynamical systems theory, the fluctuations occurring in living systems and the large number of subsystems are holistically interpreted as a complex system in the physical sense. Instead of conditioning on innumerable concrete constraints and their effects [106], DL relies on the inherent and adaptive ability of neural networks to interpolate. Thereby, the solution space is to be sampled selectively but with wide bounds [107]. Since a learner’s or patient’s body and movement coordination is constantly changing over time [108,109], and that too without intervention [110,111,112], the search for an eventual movement solution [113,114] in terms of an absolute minimum in a potential landscape can only be considered as a preliminary approach to roughly find a range of possible solutions. In this context, it does not matter whether the search strategy follows a linear slope [114] or a simulated annealing process [22]. When even the absolute minimum is constantly moving across the landscape, reliance on and training of spontaneous adjustments, as suggested by DL, seem to be of even greater importance, especially in the context of avoiding injuries that occur within the first milliseconds of landing or contact.

In summary, both intervention groups with clearly increased motoric variations in the form of a multitude of exercises led to increased improvement rates during the rehabilitation process, more than the control group with their daily routines. Based on the expected hypothesis, both intervention groups, DL and VMT, showed comparable results, with the difference that the effect size of DL was higher than VMT in most variables. The results of the studies examined provide evidence of how Differential Learning contributes to a positive increase in the performance obtained by athletes by promoting the divergent development of movement coordination and the perception and apperception of the setting [115]. Some studies have highlighted that the qualitative nature of boundary conditions is a feature of relevance that can be manipulated to promote exploratory learning [116]. Convergent guided discovery [117], as applied in the study by Behzadnia et al., can be considered an intermediate step from fully control-oriented instruction to promoting individuality through divergent self-organization. In the context of this badminton experiment, positive effects were observed on self-motivation, skill learning, and performance [118].

## 8. Limitations

Since we mainly rely on the original Fisher-statistics [119], extended by the effect sizes according to Neyman-Pearson [120], there is no claim of generalizability [121,122,123]. The scope of the study, and thus its limitations, is determined by its assumptions. Therefore, instead of limitations, aspects are discussed here that could concern obvious future questions. Firstly, the protocol of this study was retrospectively registered. Although the lack of prospective registration could have introduced possible sources of bias [54], we developed and reported this RCT following the CONSORT [124] guidelines to improve its overall methodological quality. Secondly, we only investigated the interventions on a specific population, including male handball, volleyball, and basketball players. Athletes from other types of sports activities should be investigated as well (e.g., soccer, rugby). Thirdly, other biomechanical parameters, such as muscle activities by electromyography, could provide more comprehensive data and information regarding interventions’ effectiveness. Fourthly, the investigation of the influence of daytime and individualized amount of variation of the interventions could provide further insight into the understanding of rehabilitation processes [25]. 

## 9. Conclusions

This study has shown that under the given conditions the two recently introduced motor learning approaches (DL or VMT) are superior to the traditional intervention philosophy in terms of improving the biomechanics of landing in patients after ACL reconstruction who have completed conventional postoperative rehabilitation. The DL and VMT training program can positively influence the second risk factors for ACL injury (performance, biomechanics, or psychology) and represents a serious alternative strategy that can be integrated during and after conventional postoperative rehabilitation. Therapists, coaches, trainers, and clinicians might consider using slightly modified and more open-ended instructions to promote training in their daily work when implementing neuromuscular training programs with athletes.

## Figures and Tables

**Figure 1 jcm-12-02845-f001:**
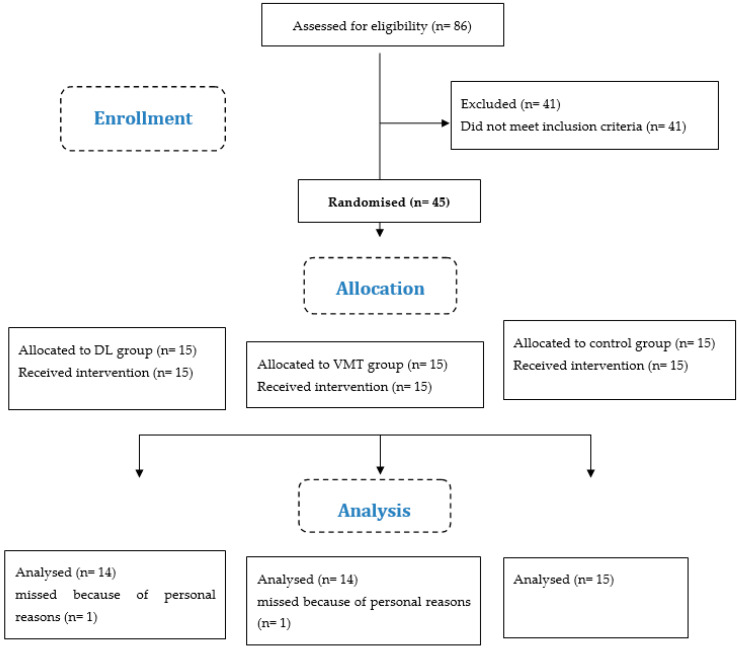
Flow diagram of the study. Abbreviations: DL: differential learning; VMT: visuo-motor training; n: number of athletes (see also Appendix A).

**Table 1 jcm-12-02845-t001:** Mean (SD) of groups’ demographic characteristics.

Groups	DL (n = 15)	VMT (n = 15)	Control (n = 15)	*p*-Value *
Age (years)	28.8 (4.6)	29.1 (3.7)	27.9 (2.9)	0.673
Mass (kg)	78.4 (6)	77 (4.8)	79 (4.9)	0.566
Height (cm)	175.2 (4.7)	174 (4.6)	174.3 (3.9)	0.710
Body mass index (kg/m^2^)	22.4 (1.9)	22.1 (1.4)	22.6 (1.5)	0.649
Time since surgery (months)	8.5 (1.1)	8.8 (1.2)	7.8 (1.5)	0.392

SD: Standard deviation, DL: Differential learning group, VMT: Visual motor training group, kg: kilogram, cm: centimeter, m: meter. * One-way analysis of variance.

**Table 2 jcm-12-02845-t002:** Differential training exercises [13].

Internal Variants:	External Variants:
*Cognition/Coordination-oriented* Before jumping: - 2–3 bunny hops - skipping both/left/right leg - high knees both/left/right knee - butt-kicks both/left/right leg - zigzag - shuffle to the left, right - complete turn to left/right before you jump While jumping: - arms crossing in front of the chest, behind the back, - raise both/left/right arm - circle both/left/right arm - head to left/right - close left/right eye While landing: - one arm in front and the other arm behind - landing with a very wide/narrow stance - landing on toes *Metabolism/Mentally oriented:* Fatigue: - With—Without	With primary stimulations of sensory system (apperception): Visual: - in a virtual reality - environment Somatosensory: - exercise on sand - with or without shoes Proprioceptive: - exercise in dark - with weight vest Acoustic: - loud music - noise from the audience

**Table 3 jcm-12-02845-t003:** Visual-motor training exercises [55].

Exercise	Visual Cues	
Tap-test	Tap the cones Acoustic Cue	The Tap-test requires the athlete to run 10 m to tap a cone, cut to the right or left for 5 m to tap another cone, cut to the opposite direction for 10 m to tap the third cone, return to the center by cutting 5 m to tap the first cone and then run 10 m back to the start position—thereby running in a “T” formation). A modification that increases the difficulty of this task and simulates the cognitive demands of sport is to have the clinician call out “Left” or “Right” to indicate which direction the athlete should cut prior to reaching the first cone, thereby creating an unanticipated cutting task which has been previously associated with increased injury-risk biomechanics compared to anticipated trials.
Agility Ladder Drills	The confines of the ladder	Agility ladder drills require athletes to match specific foot-placement patterns within an agility ladder context.
Single-leg Deadlifts	Place an object by the cone(s)	Single-leg deadlifts may be modified by requiring athletes to gently place a small object on the ground next to a cone target. To increase the difficulty, multiple cones can be placed at different angles within the athlete’s field of view, set at distances equal to his or her max volitional reaching distance while standing on one leg. For example, if the clinician chooses to use three targets, then he or she may call out “Left”, “Center”, or “Right” to vary the task order and difficulty.
Single-leg Stance (on foam)	Hold the bar horizontally	Single-leg stance on a foam surface may be modified by having the participant hold a light-weight bar with an outstretched arm and focus on keeping it steadily horizontal
Vertical Jumps	Hit the overhead target	The VERTEC is a therapeutic tool that assesses maximum vertical jump height by requiring athletes to jump and hit an overhead target. While using the VERTEC to have athletes hit the mark equal to 80% of their maximal jump height, clinicians may call out “Left” or “Right” during the initial flight phase of the jump to signal to the athlete to unilaterally land on his or her left or right leg. The use of spontaneous cuing creates an unanticipated landing task, which has been previously associated with increased injury-risk biomechanics compared to anticipated landing.
Squat Jumps	Land facing the cones	Jump squats may be modified by placing four cones around the participant at 0, 90, 180, and 270-degree positions. After numbering each cone one through four, the clinician may then rapidly call out cues to the athlete to specify which cone they should face after each jump squat. To increase the difficulty of this cognitive challenge, the clinician can introduce more cones or increase the rapidity of cuing.

Clinicians should first verify that their athlete can perform all exercises successfully before incorporating Stereoscopic glass. Then clinicians may expose their athlete to Stereoscopic glass by beginning at the easiest difficulty level (highest frequency of fluctuation between transparent and opaque states). As their athlete improves performance behaviorally, clinicians may increase Stereoscopic glass difficulty to increase the visual-cognitive demand.

**Table 4 jcm-12-02845-t004:** Error Scoring System Used to Assess Behavioral Performance [39].

Exercise	Error Count
T-test	Miss a cone Cut to the wrong direction
Agility Ladder Drills	Hit the ladder Incorrect foot placement
Single-leg Deadlifts	Opposite foot touches ground Either hand touches ground Object placed in wrong location
Single-leg Stance (on foam)	Opposite foot touches ground Either hand touches ground
Vertical Jumps	Miss the target Land on wrong foot
Squat Jumps	Land facing wrong orientation

**Table 5 jcm-12-02845-t005:** Between and within-group changes of triple hop test (cm).

Variable	Group	Pre-Test Mean ± SD	Post-Test Mean ± SD	ES (CI95%) †	*p* Value		
					Main Effect of Time	Main Effect of Group	Group × Time Interaction
Triple hop test (cm)	DL	457.3 ± 51.2	531.3 ± 74.4 ‡ §	1.15 ! (0.06 to 2.25)	F = 16.226 *p* < 0.001 *	F = 2.609 *p* < 0.085	F = 3.861 *p* < 0.029 *
	VMT	456.4 ± 54	517.6 ± 78.9 ‡	0.90 ! (−0.15 to 1.96)			
	Control	455.6 ± 51	457.4 ± 51.1	0.03 (−0.97 to 1.04)			

DL: differential learning; VMT: visual motor training; * = statistically significant difference (*p* < 0.05); ‡ = pretest to posttest significant difference; † = effect size (95% confidence intervals); ! = large Cohen’s d effect size (>0.8); Bonferroni Post Hoc test: § = significantly different from control group (*p* < 0.05).

**Table 6 jcm-12-02845-t006:** Statistical results for between and within-group changes of Star Excursion Balance Test (cm).

Variables (cm)	Group	Pre-Test Mean ± SD	8-Weeks Mean ± SD	ES (CI95%) †	*p* Value		
					Main Effect of Time	Main Effect of Group	Group × Time Interaction
Anterior	DL	80.6 ± 3.9	89.5 ± 1.9 ‡ §	2.90 ! (1.45 to 4.35)	F = 170.914 *p* < 0.001 *	F = 13.324 *p* < 0.001 *	F = 25.849 *p* < 0.001 *
	VMT	80.4 ± 3.2	88.8 ± 1.4 ‡ §	3.40 ! (1.81 to 4.98)			
	Control	80.1 ± 3	81.5 ± 3.2	0.45 (−0.57 to 1.47)			
Anteromedial	DL	81.8 ± 5	90.1 ± 3.4 ‡ §	1.94 ! (0.71 to 3.16)	F = 48.253 *p* < 0.001 *	F = 6.221 *p* < 0.004 *	F = 9.617 *p* < 0.001 *
	VMT	82.2 ± 4.8	89.8 ± 3.2 ‡ §	1.86 ! (0.65 to 3.07)			
	Control	82.5 ± 4.5	83.4 ± 3.9	0.21 (−0.80 to 1.22)			
Medial	DL	81.8 ± 5	90.1 ± 3.4 ‡	1.94 ! (0.71 to 3.16)	F = 74.274 *p* < 0.001 *	F = 3.514 *p* < 0.039 *	F = 13.337 *p* < 0.001 *
	VMT	82.2 ± 4.8	89.8 ± 3.2 ‡	1.86 ! (0.65 to 3.07)			
	Control	82.5 ± 4.5	83.4 ± 3.9	0.21 (−0.80 to 1.22)			
Posteromedial	DL	78.7 ± 3.1	89 ± 4.4 ‡ §	2.70 ! (1.30 to 4.10)	F = 162.347 *p* < 0.001 *	F = 12.205 *p* < 0.001 *	F = 31.295 *p* < 0.001 *
	VMT	78.7 ± 3.1	87.8 ± 3.2 ‡ §	2.88 ! (1.44 to 4.33)			
	Control	78.7 ± 3.1	79.6 ± 2.9	0.30 (−0.26 to 1.83)			
Posterior	DL	82.2 ± 3.3	90.1 ± 3.3 ‡ §	2.39 ! (1.06 to 3.72)	F = 84.773 *p* < 0.001 *	F = 5.316 *p* < 0.009 *	F = 16.797 *p* < 0.013 *
	VMT	81.9 ± 3.8	89.8 ± 3.3 ‡ §	2.22 ! (0.93 to 3.50)			
	Control	82.6 ± 3.7	83.2 ± 3.8	0.16 (−0.85 to 1.17)			
Posterolateral	DL	76.4 ± 3.2	87 ± 4 ‡ §	2.92 ! (1.47 to 4.38)	F = 147.187 *p* < 0.001 *	F = 10.384 *p* < 0.001 *	F = 26.928 *p* < 0.001 *
	VMT	76.6 ± 3.3	85.9 ± 4 ‡ §	2.53 ! (1.17 to 3.89)			
	Control	76.9 ± 3.3	78 ± 1.8	0.41 (−0.60 to 1.43)			
Lateral	DL	74.8 ± 4.6	87.6 ± 4.6 ‡ §	2.78 ! (1.36 to 4.20)	F = 123.620 *p* < 0.001 *	F = 9.974 *p* < 0.001 *	F = 23.937 *p* < 0.001 *
	VMT	74.3 ± 4.4	85.2 ± 3.7 ‡ §	2.68 ! (1.28 to 4.07)			
	Control	75 ± 4	76.1 ± 4.6	0.25 (−0.76 to 1.27)			
Anterolateral	DL	77 ± 4	88.6 ± 4 ‡ §	2.9 ! (1.45 to 4.35)	F = 98.265 *p* < 0.001 *	F = 10.488 *p* < 0.001 *	F = 22.583 *p* < 0.013 *
	VMT	78 ± 3.9	87.5 ± 4.1 ‡ §	2.37 ! (1.05 to 3.69)			
	Control	77.6 ± 4.3	78 ± 4.7	0.08 (−0.92 to 1.10)			

DL: differential learning; VMT: visual motor training; * = statistically significant difference (*p* < 0.05); ‡ = pretest to posttest significant difference; † = Effect size (95% confidence intervals); ! = large Cohen’s d effect size (0.8); Bonferroni Post Hoc test: § = significantly different from control group (*p* < 0.05).

**Table 7 jcm-12-02845-t007:** Statistical results for between and within-group changes of kinetic and kinematics.

Variables	Group	Pre-Test Mean ± SD	8-Weeks Mean ± SD	ES (CI95%) †	*p* Value		
					Main Effect of Time	Main Effect of Group	Group × Time Interaction
Hip flexion (degree)	DL	55.9 ± 5.5	62 ± 6.1 ‡ §	1.05 ! (−0.29 to 2.13)	F = 31.011 *p* < 0.001 *	F = 2.979 *p* < 0.062	F = 8.386 *p* < 0.001 *
	VMT	50.5 ± 7.7	56.6 ± 7.8 ‡ §	0.78 (−0.26 to 1.83)			
	Control	51.9 ± 11	51.8 ± 11	−0.00 (−1.02 to 1)			
KF (degree)	DL	28.1 ± 7.1	39.2 ± 4.5 ‡ §	1.86 ! (0.65 to 3.08)	F = 55.063 *p* < 0.001 *	F = 20.632 *p* < 0.001 *	F = 18.190 *p* < 0.001 *
	VMT	27.8 ± 5.9	41.7 ± 3.07 ‡ §	2.95 ! (1.49 to 4.42)			
	Control	26.6 ± 4.8	25.9 ± 6	−0.12 (−1.21 to 0.81)			
AD (degree)	DL	18.2 ± 1.8	23 ± 3.6 ‡	1.68 ! (−0.50 to 2.86)	F = 53.829 *p* < 0.001 *	F = 4.095 *p* < 0.024 *	F = 8.469 *p* < 0.001 *
	VMT	19.7 ± 1.2	25.7 ± 4.3 ‡	1.90 ! (0.68 to 3.12)			
	Control	21.9 ± 2.2	22.1 ± 2.2	0.09 (−0.61 to 1.43)			
KV (degree)	DL	−4.9 ± 0.2	−3.6 ± 0.2 ‡ §	6.5 ! (3.96 to 9.03)	F = 119.261 *p* < 0.001 *	F = 53.577 *p* < 0.001 *	F = 39.136 *p* < 0.001 *
	VMT	−5.3 ± 0.4	−3.8 ± 0.3 ‡ §	4.24 ! (2.41to 6.06)			
	Control	−5.1 ± 0.4	−5.2 ± 0.4	−0.12 (−1.26 to 0.76)			
VGRF (N)	DL	3.3 ± 0.2	2.8 ± 0.1 ‡ §	−3.16 ! (−4.68 to −1.64)	F = 51.717 *p* < 0.001 *	F = 8.211 *p* < 0.001 *	F = 4.870 *p* < 0.013 *
	VMT	3 ± 0.4	2.7 ± 0.1 ‡ §	−1.02 ! (−2.10 to 0.04)			
	Control	3.4 ± 0.4	3.3 ± 0.5	−0.22 (−1.52 to 0.52)			

DL: differential learning; VMT: visual motor training; * = statistically significant difference (*p* < 0.05); ‡ = pretest to posttest significant difference; † = effect size (95% confidence intervals); ! = large Cohen’s d effect size (0.8); Bonferroni Post Hoc test: § = significantly different from control group (*p* < 0.05).

**Table 8 jcm-12-02845-t008:** Between and within-group changes in TSK test.

Variable	Group	Pre-Test Mean ± SD	8-Weeks Mean ± SD	ES (CI95%) †	*p* Value		
					Main Effect of Time	Main Effect of Group	Group × Time Interaction
TSK test	DL	37.6 ± 7.3	22.6 ± 4.8 ‡ §	−2.42 ! (−3.76 to −1.09)	F = 50.047 *p* < 0.001 *	F = 3.438 *p* < 0.029 *	F = 6.154 *p* < 0.001 *
	VMT	38 ± 6.7	23.3 ± 4.1 ‡ §	−2.64 ! (−4.03 to −1.26)			
	Control	37.4 ± 6.5	36.2 ± 6.0	−0.19 (−1.20 to 0.82)			

DL: differential learning; VMT: visual motor training; * = statistically significant difference (*p* < 0.05); ‡ = pretest to posttest significant difference; † = Effect size (95% confidence intervals); ! = large Cohen’s d effect size (0.8); Bonferroni Post Hoc test: § = significantly different from control group (*p* < 0.05).

## Data Availability

The datasets generated during and/or analyzed during the current study are available from the corresponding author on reasonable request.

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
