# Peer review of "Comparing the Effects of Differential and Visuo-Motor Training on Functional Performance, Biomechanical, and Psychological Factors in Athletes after ACL Reconstruction: A Randomized Controlled Trial"

_jcm, 2023, doi:10.3390/jcm12082845_

Round 1

Reviewer 1 Report

Dear Authors!

First at all I would like to congratulate you on a well-conducted study. It was a pleasure to read your article. In accordance with the reviewer,s obligation I would like to propose a few corrections in the text.

1. Section "Results" - You repeat the same informations in the introduction to each table and in the corresponding table. Write in introduction one or two general phrases and use formula -"... particular results are shown in table below".

2. Section "Discussion" - I would like to suggest  deletion some phrases (lines from 398 to 410) because they contain information presented in section "Results".

With regards

Author Response

Authors’ response:

Dear Editor,

Thank you for the possibility you have offered us to continue to improve the quality of the manuscript taking into consideration the criticism and the feedback of experts in the field.

We are happy that reviewers’ output led to the fewer comments and critics.

#Reviewer 1

Dear Authors!

First at all I would like to congratulate you on a well-conducted study. It was a pleasure to read your article. In accordance with the reviewer,s obligation I would like to propose a few corrections in the text.

Dear Reviewer,

Thanks for your interest in our manuscript. Your appreciation of our review and your positive feedback is appreciated. We have carefully considered your valuable comments and revised its contents in depth.

  1. Section "Results" - You repeat the same informations in the introduction to each table and in the corresponding table. Write in introduction one or two general phrases and use formula -"... particular results are shown in table below".

Answer: duplicate items were removed in the result section.

  1. Section "Discussion" - I would like to suggest deletion some phrases (lines from 398 to 410) because they contain information presented in section "Results".

Answer: thanks for your valuable comment, Since the main findings of the study are given at the beginning of the discussion and then these main findings are discussed, so removing these items will confuse the reader.

Reviewer 2 Report

Dear author, 

I am pleased to submit to you my review of your article. 

The topic is interesting, current, and relevant to our clinical practice. 

The article is well written, but many concerns burden it with a minor revision.  Congratulations to the authors for their efforts.

Some suggested corrections have been listed below. Please answer point by point.

-Write the numbers in the manuscript in the same way. Spell out numbers one through nine, except for units of measure or time. For values of 10 and higher, use Arabic numerals. Always spell out numbers at the beginning of a sentence if the sentence cannot be rearranged to avoid starting with a number.

-The Introduction is too long. A general rule is that the Introduction is approximately one manuscript page long. The Introduction should contain some information about why this study is needed, which gap in the literature you would like to fill, and the purpose and hypothesis.

-Please provide the exact hypothesis in detail when explaining the purpose of the study. The reader must also understand why your study is clinically relevant. What makes your analysis more appropriate than others already published? What is new about it? What makes your study necessary? What is different about it? What is comparable? You must indicate what is original about your research. What makes it worthy of publication?

-Start the discussions by affirming the study's main finding, to be added at the beginning of the section.

-At the end, please mention the clinical relevance of your work. How can this work be valuable in day-by-day clinical work? Add the study's strengths, which are essential to a high-quality written manuscript. Furthermore, the study's limitations should be analyzed in more detail.

-Line 177: At the end of the sentence "based on biomechanical and joint position sense data", I suggest including recent references that mention the importance of ACL biomechanics. I recommend you add these related references: doi: 10.1007/s00402-021-04030-8; doi: 10.1016/j.jor.2022.11.018; doi: 10.1007/s00167-013-2449-4.

-Line 203: Add the abbreviation for "n," for example, n: number of evaluation cases.

-Line 433-434: -It would be appropriate to expand the bibliography. Other recent studies specifically analyze the topic of varus-valgus deformity and its relationship with the risk of ACL injury. I suggest you add these related references at the end of the sentence: doi: 10.1136/bjsports-2015-094776; doi: 10.21037/aoj-22-1; doi: 10.1007/s00590-022-03419-4.

- TABLES: Your measurement methods must be given in detail. Measurement accuracy is necessary to report. Please ensure your results are given with the same accuracy as the methods. If your methods allow one decimal, the result should also be reported with one decimal. Information about measurement accuracy is essential.

-Please keep repetitions to a minimum. Unfortunately, there are several repetitions in information in the Results section and Tables.

Author Response

#Reviewer 2

Dear author, 

I am pleased to submit to you my review of your article. 

The topic is interesting, current, and relevant to our clinical practice. 

The article is well written, but many concerns burden it with a minor revision.  Congratulations to the authors for their efforts.

Some suggested corrections have been listed below. Please answer point by point.

-Write the numbers in the manuscript in the same way. Spell out numbers one through nine, except for units of measure or time. For values of 10 and higher, use Arabic numerals. Always spell out numbers at the beginning of a sentence if the sentence cannot be rearranged to avoid starting with a number.

 Answer: As you suggested, we have implemented all the requested changes. Thanks.

-The Introduction is too long. A general rule is that the Introduction is approximately one manuscript page long. The Introduction should contain some information about why this study is needed, which gap in the literature you would like to fill, and the purpose and hypothesis.

Answer: The introduction has been shortened. From a further shortening of the introduction, we abstained because of the explanations of the new applied approaches including their differences that may be not known by most of the readers.

-Please provide the exact hypothesis in detail when explaining the purpose of the study. The reader must also understand why your study is clinically relevant. What makes your analysis more appropriate than others already published? What is new about it? What makes your study necessary? What is different about it? What is comparable? You must indicate what is original about your research. What makes it worthy of publication?

Answer: The hypotheses are provided at the end of the introduction (line 296-298). The introduction was reformulated at certain passages to make the clinical relevance clearer. Similarly, we hope that due to the removing of some information and reformulating some passages it may become clearer what is new/different in comparison to the old. What is compared is described in line 294 ff (, this study aimed to …)  based on the previous outlining of the existing and new approaches. The application of the new approaches and their superiority in comparison to the traditional approach makes it worth to be published.

-Start the discussions by affirming the study's main finding, to be added at the beginning of the section.

Answer: the main findings of the study are given at the beginning of the discussion, then according to the order of the variables in the results section, first the findings of that variable and then the discussion about it are given.

-At the end, please mention the clinical relevance of your work. How can this work be valuable in day-by-day clinical work? Add the study's strengths, which are essential to a high-quality written manuscript. Furthermore, the study's limitations should be analyzed in more detail.

Response: The conclusions have been reformulated.

The limitations were already formulated in the chapter "Perspectives", but with a more differentiated epistemological background referring to the current replication crisis (RC). The RC mainly complains about false generalizations and, consequently, leads to a low replication rate due to only slightly changed boundary conditions in living systems described by the Duhem-Quine theorem. The limitations of the study arise from the details of the boundary conditions described in the methods section including the choice of the design. Instead of repeating the methods section, we decided to reflect on the next steps that should be the subject of the research based on the chosen boundary conditions and the results. To make the reader aware of this problem, the section was renamed "limitations" instead of "Perspectives".  

-Line 177: At the end of the sentence "based on biomechanical and joint position sense data", I suggest including recent references that mention the importance of ACL biomechanics. I recommend you add these related references: doi: 10.1007/s00402-021-04030-8; doi: 10.1016/j.jor.2022.11.018; doi: 10.1007/s00167-013-2449-4.

Answer: thanks to dear reviewer, recommended references were added to the desired sentence.

-Line 203: Add the abbreviation for "n," for example, n: number of evaluation cases.

Answer: done.

-Line 433-434: -It would be appropriate to expand the bibliography. Other recent studies specifically analyze the topic of varus-valgus deformity and its relationship with the risk of ACL injury. I suggest you add these related references at the end of the sentence: doi: 10.1136/bjsports-2015-094776; doi: 10.21037/aoj-22-1; doi: 10.1007/s00590-022-03419-4.

Answer: really thanks to reviewer, the second reference (doi: 10.21037/aoj-22-1) information was not correct, two suggested references were added to the relevant section.

- TABLES: Your measurement methods must be given in detail. Measurement accuracy is necessary to report. Please ensure your results are given with the same accuracy as the methods. If your methods allow one decimal, the result should also be reported with one decimal. Information about measurement accuracy is essential.

Answer: the mean and standard deviation of all variables and demographic data were reported with one decimal.

-Please keep repetitions to a minimum. Unfortunately, there are several repetitions in information in the Results section and Tables.

Answer: Duplicate items were removed from the title of the variables in the results section.